# Effect of Fermentation Time on Antioxidant and Anti-Ageing Properties of Green Coffee Kombucha Ferments

**DOI:** 10.3390/molecules25225394

**Published:** 2020-11-18

**Authors:** Nizioł-Łukaszewska Zofia, Ziemlewska Aleksandra, Bujak Tomasz, Zagórska-Dziok Martyna, Zarębska Magdalena, Hordyjewicz-Baran Zofia, Wasilewski Tomasz

**Affiliations:** 1Department of Technology of Cosmetic and Pharmaceutical Products, Medical College, University of Information Technology and Management in Rzeszow, Kielnarowa 386a, 36-020 Tyczyn, Poland; znizol@wsiz.rzeszow.pl (N.-Ł.Z.); aziemlewska@wsiz.rzeszow.pl (Z.A.); martynazd89@gmail.com (Z.-D.M.); 2ŁUKASIEWICZ Research Network—Institute of Heavy Organic Synthesis “Blachownia”, Energetykow 9, 47-225 Kedzierzyn-Kozle, Poland; magdalena.zarebska@icso.lukasiewicz.gov.pl (Z.M.); zofia.hordyjewicz@icso.lukasiewicz.gov.pl (H.-B.Z.); 3Department of Industrial Chemistry, University of Technology and Humanities in Radom, Chrobrego 27, 26-600 Radom, Poland; tomasz.wasilewski@uthrad.pl; 4Research and Development Department, ONLYBIO.life Sp. z o.o., Wojska Polskiego 65, 85-825 Bydgoszcz, Poland

**Keywords:** kombucha, green coffee, ferment, antioxidant activity, matrix metallopeptidases

## Abstract

Kombucha, also known as the Manchurian mushroom, is a symbiotic culture of bacteria and yeast, the so-called SCOBY. This paper presents a comprehensive evaluation of the ferments obtained from green coffee beans after different fermentation times with kombucha. Results for the ferments were compared to the green coffee extract that was not fermented. In this study, the antioxidant potential of obtained ferments was analyzed by assessing the scavenging of external and intracellular free radicals and the assessment of superoxide dismutase activity. Cytotoxicity of ferments on keratinocyte and fibroblast cell lines was assessed as well as anti-aging properties by determining their ability to inhibit the activity of collagenase and elastase enzymes. In addition, the composition of the obtained ferments and the extract was determined, as well as their influence on skin hydration and transepidermal water loss (TEWL) after application of samples on the skin. It has been shown that the fermentation time has a positive effect on the content of bioactive compounds and antioxidant properties. The highest values were recorded for the tested samples after 28 days of fermentation. After 14 days of the fermentation process, it was observed that the analyzed ferments were characterized by low cytotoxicity to keratinocytes and fibroblasts. On the other hand, the short fermentation time of 7 days had a negative effect on the properties of the analyzed ferments. The obtained results indicate that both green coffee extracts and ferments can be an innovative ingredient of cosmetic products.

## 1. Introduction

Scientists around the world are looking for new compounds of natural origin that show a beneficial effect on human health. Due to the fact that it has been known for a long time that various biochemical transformations of plant raw materials can increase their health value, various methods of obtaining new substances with desired properties are sought. Fermentation is one of the methods of processing plant raw materials, causing many changes that lead to obtaining valuable bioproducts with high nutritional value. Thanks to its application, it is possible to enrich the obtained products by improving their bioavailability, degradation of toxic ingredients and anti-nutritional factors, improving sensory quality as well as giving them additional medical and therapeutic properties [1]. Fermentation is a bioprocess widely used for the production of biologically active compounds, primarily in the food industry, but recently, it is increasingly used in cosmetics to improve the quality of active phytochemicals and to facilitate the absorption of these substances by the human system [2]. Extracts obtained during food and beverages’ fermentation are a rich source of various compounds with antioxidant properties, vitamins, minerals, proteins, as well as fibers and probiotics, hence the interest in this type of bioproducts is constantly increasing. Therefore, research is currently underway to assess the possibilities of their use in the cosmetics industry as products with strong antioxidant, anti-inflammatory, anti-wrinkle, whitening and anti-aging properties [2,3]. Interest in ferments in the design of cosmetic formulation is increasing due to their documented ability to inhibit matrix metalloproteinases, stimulate collagen production, enhance skin hydration, which is important in skin aging processes, and to inhibit tyrosinase activity, which protects against formation of various hyperpigmentation changes [2,3,4,5,6].

One of the “health trends” that have recently gained great popularity are ferments obtained using the symbiotic culture of bacteria and yeast, called SCOBY. Ferments obtained using this microbiological consortium of several bacteria and yeast, called kombucha, quickly found a wide range of consumers due to their unusual properties. Fermentation products are obtained thanks to symbiotic cooperation of the community of acetic acid bacteria (*Acetobacteraceae*) and osmophilic yeast [7,8]. Although the “kombucha” refers mainly to ferments made from various types of tea, currently, attempts are being made to obtain bioferments using SCOBY also from other plant materials such as, for example, mint, lemon balm, jasmine, spinach, fermented fruit juices, banana peels, milk or even bee-collected pollen [7,9,10]. Scientific research on kombucha properties has shown that it exhibits antioxidant, antibacterial, anti-inflammatory and antidiabetic properties. Consumption of kombucha can also contribute to lowering cholesterol, stimulating liver detoxification processes and also supporting the proper functioning of the immune system. Moreover, kombucha is also a rich source of various vitamins, phenolic compounds, minerals, amino acids and many other compounds with a broad spectrum of biological activity [8,9,10,11,12,13].

In addition to tea, one of the most popular beverages consumed around the world is coffee, including green coffee. The biological active substances contained in coffee extracts include two groups of compounds. The first are substances with antioxidant activity, such as polyphenols, which allow to neutralize various forms of superoxide free radicals. For this reason, they are particularly valuable in protecting the DNA of skin cells. The second group are purine alkaloids, of which caffeine and trigonelline are the most important. These compounds improve cell oxygenation and microcirculation and stimulate metabolism. Many literature reports indicate its extraordinary properties, which contribute to the fact that extracts obtained from this plant have been widely used for many years [14,15,16]. Due to the fact that the preparation of infusions from green coffee is similar to the preparation of tea, perhaps fermentation using SCOBY will contribute to obtaining beverages with extremely valuable properties. Unfortunately, to date, there are few popular science reports indicating the beneficial effects of coffee bioferments [17], while there is a lack of scientific reports on green coffee ferments.

Therefore, the purpose of this work is to evaluate the properties of ferments obtained from green coffee beans after fermentation using cooperation in complex multi-species systems and investigate the impact of fermentation time on the biological activity of the obtained products. For this purpose, the composition of prepared ferments was determined using a chromatographic method and a spectrophotometric evaluation of flavonoids and phenols content was carried out. The antioxidant properties of prepared ferments were also estimated by evaluating the scavenging of extrinsic and intracellular free radicals, as well as the assessment of superoxide dismutase activity. In order to assess anti-aging properties, the possibility of inhibiting the activity of matrix metalloproteinases, collagenase and elastase, which play an important role in skin aging processes, was evaluated. Cytotoxicity tests (Alamar Blue, Neutral Red and lactic dehydrogenase (LDH)) were also carried out on skin cells, keratinocytes and fibroblasts, which allowed assessment of the cytotoxicity of prepared ferments. In the final stage, the influence of the obtained ferments on skin moisture and transdermal water loss was also evaluated.

## 2. Results and Discussion

### 2.1. Determination of Bioactive Compounds

Polyphenols and flavonoids are the basic active ingredients of plant extracts. They are responsible for their antioxidant activity by neutralizing free radicals that may generate oxidative stress. Green coffee is a rich source of polyphenols and flavonoids. Depending on the origin, green coffee beans contain about 6–10% of polyphenols. The main polyphenol compounds in green coffee beans are chlorogenic acids. They may be present in concentrations up to 90% of the total phenolic content. Green coffee also contains a high concentration of caffeine derivatives, which can range from 1% to 5% [18,19].

The content of phenolic compounds present in kombucha ferments and green coffee extracts was spectrophotometrically and chromatographically determined using the Folin–Ciocalteu assay and HPLC-UV-ESI-MS, respectively. In spite of the accepted un-specificity of the Folin–Ciocalteu assay to determine phenolic compounds, it is the most commonly used method to estimate phenolic contents. The phenolic contents in the obtained kombucha ferments (F7, F14, F28) and green coffee beans’ extract are presented in Table 1.

Fermentation of green coffee extract reduces the content of polyphenols and flavonoids in relation to the content of these compounds in the unfermented extract. The lowest content of these components was observed for 7-day ferment (total phenolic content (TPC) 106.76 ± 3.72 mg GAE/g DW and total flavonoids content (TFC) 17.02 ± 2.33 mg QE/g DW). With the increase of fermentation time, TPC and TFC increased. After 14 days of fermentation, the content of the analyzed compounds increased more than twice (TPC 243.14 ± 3.22 mg GAE/g DW and TFC 51.1 ± 2.84 mg QE/g DW). The highest content of polyphenols and flavonoids was observed after 28 days of fermentation. TPC was 392.56 ± 1.22 mg GAE/g and TFC increased to 66.53 ± 3.84 mg QE/g DW. However, these values were lower than determined for the green coffee extract (TPC 630.05 ± 5.20 mg GAE/g DW and TFC 156.84 ± 4.11 mg QE/g DW).

Chu and Chen [20] have shown that change in the content of polyphenols and flavonoids during the fermentation process may result from the polymerization of these compounds that occur at the beginning of the fermentation. Polymerized compounds with high molecular weight reduce the detected polyphenol content. The increase in the concentration of polyphenolic compounds with longer fermentation time may be caused by depolymerization of the polymerized active compounds. This thesis can be confirmed by organoleptic observation of changes occurring during the fermentation process. Initially, the clear ferment becomes cloudy after about 7 days. This could have been due to the appearance of polymerized compounds with high molecular weight and limited water solubility. After about 14 days, the cloudy ferment becomes opalescent and clear in the following days. This could have been caused by depolymerization and the appearance of compounds with higher solubility.

Simultaneously, a chromatographic method was developed to deepen the chemical structures of the active compounds and, in addition, accurately determine their content.

The identification of active compounds was performed using the combined data of elution order on reverse phase, co-chromatography with standards, characteristics of diode-array detector (DAD), as compared to standards analyzed under the same conditions. The compounds, for which there were no commercial standards available, were identified based on the elution order on reverse phase [21,22,23] and ESI-MS analysis. Table 2 includes active compounds detected using HPLC-UV-ESI-MS.

The most important bioactive compounds of coffee include phenolic compounds (such as chlorogenic acids and derivatives), alkaloid from methylxanthine group (caffeine), nicotinic acid (vitamin B3) and its precursor trigonelline, magnesium and potassium [24]. The main groups of phenolic compounds in coffee bean extracts are chlorogenic acids (CGA), which are esters of quinic acid and trans-hydroxy cinnamic acid. Various chlorogenic acid derivatives arise from isomers in the quinic acid part and substitutions in the cinnamic acid moiety. The obtained results of HPLC-UV-ESI-MS for kombucha ferments (F7, F14, F28) and green coffee beans’ extracts revealed the presence of caffeine and trigonelline in positive-ion mode and phenolic compounds in negative-ion mode. Phenolic compounds having appropriate molecular ion were assigned as caffeic acid (*m*/*z* 178), quinic acid (*m*/*z* 191) with two isomers, 5-coumaroylquinic acid (*m*/*z* 337), caffeoylquinic acids (*m*/*z* 353) with 3 isomers (3-,4- and 5-CQA), feruloylquinic acids (*m*/*z* 367) with 3 isomers (3-,4- and 5-FQA), dicaffeoylquinic acids (*m*/*z* 515) with 3 isomers (3,4-diCQA; 3,5-diCQA; 4,5-diCQA) and 3-Caffeoy-l,5-feruloylquinic acid (529 *m*/*z*). In this paper, International Union of Pure and Applied Chemistry (IUPAC) nomenclature and recommended numbering systems were used for CGAs. The extracted ion chromatograms obtained in negative-ion mode for coffee beans’ extract and kombucha ferments are presented in Figure 1.

CGAs, caffeine and trigonelline were quantified from the HPLC-UV chromatographic peak areas. All quantitative analyses were performed in triplicate and the results were expressed in mg/100 g of dry weight. The method showed good precision with relative standard deviation (SD) values below 5% for all the determined CGAs. The obtained results are presented in Table 3.

Total quantified CGAs contents was 6677 mg/100 g of dry weight of green coffee beans, while the individual components were CQAs 5905, diCQAs 206 and FQAs 565 mg/100 g of dry weight, which correspond to data presented by Farah and Donangelo [25]. The average total CGAs in kombucha ferments F7, F14 and F28 were 3850, 4229 and 4161 mg/100 g of dry weight, respectively. The sum of all quantified active compounds was 9839, 6176, 6304 and 6350 mg/100 g of dry coffee for GC, F7, F14 and F28, respectively.

The total caffeine, trigonelline and phenolic content determined chromatographically varied among the analyzed extracts. As indicated, green coffee beans’ extract showed the highest caffeine, trigonelline and phenolic content. The lowest active compound content was determined for F7. This was in agreement with the trend observed using the Folin–Ciocalteu assay.

In summary, caffeine, trigonelline, phenolic compounds and their derivatives have been identified in green coffee beans’ extract and kombucha ferments using HPLC-UV-ESI-MS. Green coffee beans’ extract showed higher polyphenol, caffeine and trigonelline contents than kombucha ferments. Green coffee beans’ extract represents an important source of polyphenols and alkaloids, with high antioxidant capacity.

### 2.2. 1,1-diphenyl-2-picrylhydrazyl (DPPH) Radical Scavenging Assay

Reactive oxygen species (ROS), their sources, functions and influence on the skin are the subject of numerous studies. The balance between the rate of free radical production and the concentration of antioxidants, as well as the activity of protective enzymes neutralizing ROS, determines the level of reactive oxygen forming in the body and the speed of their reaction with cell components. Free radicals are essential for the proper conduct of many life processes. They play an important role in the regulation of gene expression, protein phosphorylation processes and the activation of proteins that control cell division. However, the excess of free radicals leads to the destruction of structural and functional elements of cells, disturbances in homeostasis and death through apoptosis or necrosis [22,26,27]. That is why it is so important to search for substances being an effective source of antioxidants. Plant raw materials play a very important role here, as they are a valuable source of metabolite derivatives.

The green coffee extract can be regarded as such raw materials [22,27]. In the first stage of the research, the antioxidant potential of the analyzed green coffee extract and kombucha ferments was assessed. It was shown that the tested material exhibits a significant antioxidant potential, and the fermentation process significantly influences the percentage of free radicals scavenged by the analyzed extracts. In the case of all analyzed extracts, an increase in the antioxidant potential in time was observed. Green coffee extract showed the highest antioxidant potential. However, green coffee extract subjected to a 28-day fermentation process was also characterized by strong antioxidant properties, as after 30 min of measurement, it had only 14% lower antioxidant potential compared to the results obtained for the green coffee extract. The biggest difference was observed between the green coffee extract and the extract subjected to a 7-day fermentation process. The green coffee extract showed 120% higher potential than the ferments obtained after 7 days. A dependence between the content of active ingredients such as phenols and flavonoids and the antioxidant potential has also been demonstrated (Figure 2).

An increase in the antioxidant potential with an increase in fermentation time was also observed in the case of tea fermentation [12,27,28,29,30,31,32,33,34,35,36,37,38,39]. The conducted research shows that enzymes released by bacteria and yeasts during long-term fermentation of kombucha may result in better efficiency in scavenging nitrogen and superoxide radicals and poor efficiency in scavenging hydroxyl radicals [20,30,31].

### 2.3. Detection of Intracellular Levels of Reactive Oxygen Species (ROS)

In order to comprehensively assess the antioxidant potential, the ability of the tested ferments to generate intracellular production of reactive oxygen forms on keratinocytes (HaCaT) and fibroblast (BJ) cell lines was evaluated. These analyses were performed with the fluorogenic 2’,7’-dichlorodihydrofluorescein diacetate (H_2_DCFDA) dye, which is oxidized in the presence of reactive oxygen forms and converted into highly fluorescent 2′,7′-dichlorofluorescein (DCF). The performed analysis showed that the potential of the tested ferments was dependent on both the concentration used and the analyzed cell line.

The analysis showed that the concentration of 1000 µg/mL for both keratinocytes and fibroblasts increased the production of ROS. The result for keratinocytes was much higher, both for the extract itself and for all samples subjected to different extraction times. In the case of a concentration of 1000 µg/mL, the level of ROS in each tested case was much higher compared to the control, and the difference was increasing with time. In the case of fibroblasts, up to the 7th day, the ROS level was slightly higher compared to the control, while from the 14th day, a significant increase in the ROS level was observed. In the case of keratinocytes from the beginning of the experiment, a noticeably higher level of ROS was found, while from the 14th day, the difference was even greater.

At a concentration of 100 µg/mL, since the beginning of the experiment until the 14th day, the level of ROS for fibroblasts was slightly below the control level, while for keratinocytes, it was at the level of control. The situation changed in the case of extracts subjected to 28-day fermentation, where the level for fibroblasts increased to the control value, while for keratinocytes, it was slightly above the control level (Figure 3, Figure 4, Figure 5 and Figure 6).

Based on the analysis, it can be concluded that lower concentrations of ferments and coffee extract influence on a reduction of the level of oxidative stress. On the other hand, the long-term fermentation time (28 days) adversely affects the production of ROS. These data correspond with the thesis that too-long fermentation may not be beneficial due to the fact it may contribute to the accumulation of harmful products, e.g., organic acids, which might reach harmful levels for direct consumption [20,32].

### 2.4. Determination of Superoxide Dismutase (SOD) Activity

Prevention of threats caused by reactive oxygen species can take place at many levels, but the first line of defense is the action of ROS neutralizing enzymes. These enzymes include superoxide dismutase (SOD), which catalyzes the dismutation reaction of the superoxide radical anion to hydrogen peroxide and molecular oxygen. This reaction occurs due to the interaction of the radical anion with metal ions present in the catalytic center of the enzyme. SOD constitutes a very important antioxidant protection of the organism against oxidative stress, and their incorrect operation can lead to many pathologies. SOD is a natural enzyme present in cells and is responsible for the neutralization of harmful free radicals in order to protect DNA during intense environmental stress, being considered as an anti-aging enzyme [33,34,35]. Therefore, in order to comprehensively assess the antioxidant potential of the analyzed extracts, the level of superoxide dismutase in the tested samples was determined. As a result of the analysis, it was shown that the tested 14-day ferments at the concentration of 500 and 1000 µg/mL show a higher activity of the superoxide dismutase enzyme compared to the green coffee extract. In the case of the lowest analyzed concentration of 100 µg/mL, the green coffee extract showed the highest values. On the other hand, the ferments obtained after 7 and 28 days of fermentation were characterized by much lower values compared to the coffee extract and ferment obtained after 14 days of fermentation (Figure 7).

### 2.5. Cell Viability Assay

In the next stage of our research, we wanted to investigate how fermentation time affects the viability of skin cells; therefore, the cytotoxicity of kombucha ferments obtained from green coffee beans against fibroblasts and keratinocytes was evaluated in vitro using three types of assays (Alamar Blue test, Neutral Red uptake assay and LDH cytotoxicity test). The Alamar Blue assay is a commonly used cytotoxicity test with many biomedical applications [39]. This test contains the nonfluorescent resazurin as a primary constituent, which is reduced to the fluorescent resorufin by different oxidoreductases that use nitrate reductases (NAD(P)H) as a primary electron donor. Resazurin, as a redox indicator, is used in assays for cell proliferation, cell viability and mitochondrial respiratory activity [37,38]. Neutral Red uptake assay has already been used for several years to evaluate the cytotoxicity of various kinds of products, such as cosmetics, pharmaceuticals, industrial chemicals and household products [39]. This assay relies on the ability of living cells to incorporate and bind neutral red dye in lysosomes. This cationic dye can penetrate cell membranes by nonionic passive diffusion and bind by electrostatic hydrophobic bonds to anionic and phosphate groups of the lysosomal matrix. The uptake of this dye depends on the cells’ capacity to maintain pH gradients, which is closely related to the production of adenosine triphosphate (ATP) [40,41].

The analysis carried out to assess the cytotoxic properties of the kombucha ferments obtained from green coffee beans at tested concentrations (10–1000 μg/mL) have shown that these ferments have different effects on the tested skin cell lines. Our analysis has shown that kombucha ferments are similar in cytotoxicity, both for keratinocytes and fibroblasts (Figure 8 and Figure 9). The conducted analysis clearly indicates that the effect exerted by the tested ferment is strictly dependent on the fermentation time. The highest increase in both keratinocytes’ and fibroblasts’ proliferation was observed for ferments after 14 days of fermentation, and at the concentration of 250 and 500 μg/mL, the increase even reaches 115%. Moreover, by carrying out fermentation for a long time, it has been shown that kombucha ferments have no beneficial effect at any of the concentrations used. It is noticeable that all tested ferments after 28 days of fermentation slightly reduced the proliferation and viability of skin cells, even up to 25% for keratinocytes in the Neutral Red test, which indicates their cytotoxic effect associated with the reduction of cellular metabolism.

The cytotoxic effect of the obtained ferments was also assessed using the lactate dehydrogenase (LDH) cytotoxicity assay. The assay quantitatively measures a stable cytosolic enzyme, LDH, which is released upon cell lysis. The released LDH is measured with a coupled enzymatic reaction that results in the conversion of a tetrazolium salt (INT) into a red color formazan. The method requires monitoring of the increase in UV absorbance due to the reduction of NAD+ [42,43]. The cytotoxicity analysis carried out indicates that the effect exerted by the tested ferments also closely depends on the fermentation time. The cell viability results indicate that kombucha ferments are not toxic to fibroblasts and keratinocytes after 14 days of fermentation and there were no changes in extracellular LDH levels after exposure to ferments compared to the control culture (Figure 10). After 28 days of fermentation, membrane damage (LDH release) was observed (107% for keratinocytes at the concentration of 500 μg/mL). A correlation between extract concentration and LDH release was observed in both analyzed skin cells.

Cytotoxicity studies were compared to green coffee extract. Many studies indicate that green coffee has beneficial effects on skin cells [16,44]. In addition, green coffee extract could increase wound contraction and accelerate wound healing [44,45]. Due to the small amount of research on the fermentation of green coffee beans with kombucha, we wanted to check the effect of the obtained ferments on skin cells. Despite the many proven health benefits of SCOBY [46,47], there is still little research into the effects of kombucha on skin cells. Fontana et al. [48] have shown that the tea fungus can be used for medical purposes in skin therapy. The cellulosic pellicle formed mainly by *Acetobacter xylinum* during the fermentation of tea has been used as a temporary skin substitute on burns and in other skin injuries [48]. Our research has shown that green coffee is not as good a raw material for fermentation with kombucha as green or black tea.

### 2.6. Assessment of Matrix Metallopeptidases Inhibition

Collagen and elastin are basic skin building proteins. They are responsible for the skin’s elasticity and hydration. Age and many internal and external factors, such as ultraviolet radiation, can significantly accelerate the collagen and elastin degradation process in the skin. Protein degradation processes are stimulated by collagenase and elastase enzymes. Increases in their activity is observed as a result of action of free radicals or the mentioned UV radiation. Many plants and plant extracts have the ability to inhibit collagenase and elastase activity. This property affects, among others, the acceleration of skin regeneration processes, the healing of wounds and scars and the slowing down of skin aging processes. External use of cosmetic and pharmaceutical products containing ingredients capable of collagenase and elastase activity can contribute to slowing down of the degradation of collagen and elastin contained in the skin and stimulate their synthesis [49,50]. The ability to inhibit collagenase and elastase was determined for green coffee extract and its ferments. The results are shown in Figure 11 and Figure 12.

Green coffee extract showed the ability to inhibit both collagenase and elastase. Stronger properties were noted in relation to the first of them. In addition, it has been shown that the ability to inhibit enzymatic activity depends on the extract concentration. At an extract concentration of 100 µg/mL, collagenase inhibition was about 10%, while at a concentration of 1000 µg/mL, collagenase inhibition was about 55%. Lower values were obtained for elastase inhibition. At a concentration of 1000 µg/mL, green coffee extract showed an ability of elastase activity inhibition at a level of about 25%. As a result of research, it has been shown that green coffee ferments also have enzyme inhibition properties, but at a significantly lower level than unfermented extract. The effect of fermentation time on the ability to inhibit enzymatic activity was noted. The most favorable properties were observed for the ferment obtained in a 14-day fermentation process, for which at a concentration of 1000 µg/mL the ability for collagenase activity inhibition was determined at a level of about 30%. For other ferments, collagenase inhibition did not exceed 10–15%. In the case of the ferment obtained after 14 days, the elastase inhibition capacity was comparable to that of the green coffee extract. Other green coffee ferments were characterized by a low ability to inhibit elastase at the level of 1–5%.

Ability for metallopeptidases inhibition may be shown by active ingredients of plants. Green coffee beans are a rich source of polyphenols and flavonoids, which have strong antioxidant activity and are able to inhibit the activity of collagenase and elastase. The same properties are shown by main active ingredients of green coffee like caffeine and its derivatives, trigonelline and chlorogenic acid. Lee et al. showed that caffeine at a concentration of 500–1000 µg/mL inhibits collagenase and elastase activity at a level of 30–40% [51]. The obtained results of the metallopeptidases inhibition confirm that the concentration of active ingredients in the green coffee extract and ferments (polyphenols, flavonoids, caffeine) exerts a significant influence on their ability of collagenase and elastase activity inhibition. The highest concentration of active ingredients was shown by the green coffee extract and it also showed the highest inhibitory properties. Along with the increase in the fermentation time of coffee beans, an increase in the concentration of active ingredients was noted, as well as a greater ability to inhibit the activity of the analyzed enzymes.

### 2.7. Transepidermal Water Loss (TEWL) and Skin Moisture

For several years, plant ferments have gained increasing interest in the cosmetics industry. They are a source of many active ingredients with a wide spectrum of effective action. Due to the content of simple chemicals with low molecular weight, plant ferments are a source of bioavailable active substances that are characterized by a high degree of penetration into the deeper layers of the skin [52,53,54,55,56]. As part of this work, the effect of green coffee ferments on basic skin parameters such as skin hydration and TEWL was determined. Results are presented in Figure 13 and Figure 14.

Application of the green coffee ferments on the skin improves both skin hydration and TEWL level. After application of the analyzed substances on the skin, a significant decrease in TEWL in relation to the control field was noted. The ability of ferments to reduce TEWL was characterized by long-term effects. However, no significant effect of fermentation time on the obtained results was observed. After 1 h of the ferments’ application on the skin, a decrease in TEWL of about 7% for the 7-day ferment and about 15% for the 28-day ferment was noted. After 6 h, the decrease in TEWL remained at a high level and was lower by about 25–30% in relation to the control field. Obtained results were slightly different from results observed after application of green coffee extract on the skin. Green coffee ferments also significantly improve the skin hydration. The moisturizing effect, similarly, to lowering of TEWL, was long-lasting and showed an increasing tendency with time after the application of the analyzed samples on the forearm skin. Fermentation time of coffee beans has a little effect on the moisturizing effect of the obtained ferments. The most favorable properties were observed for the 14-day ferment. On the other hand, the 7-day ferment was characterized by the lowest moisturizing properties. For the 28-day ferment, obtained results were slightly lower than for the ferment obtained as a result of the 14-day fermentation process. The moisturizing effect increased significantly over time after application of the analyzed ferments on the skin. After 6 h of the ferments’ application on the skin, the hydration level was about 30% higher for the 7- and 28-day ferments and about 40% higher for the 14-day ferment. For this ferment, significantly higher skin hydration properties were observed than for the unfermented green coffee extract.

As was mentioned earlier, the main advantage of plant ferments is a profile of active ingredients that they contain. The most important include simple sugars, amino acids, vitamins, as well as substances from the group of polyphenols and flavonoids. As indicated by numerous literature data, as a result of fermentation processes, complex organic compounds (e.g., sugars and proteins) are broken down into simple substances such as glucose, fructose or amino acids [7,11,20,28]. These substances are characterized by a lower molecular weight and smaller particle sizes, which may affect their higher ability to penetrate into deeper layers of the epidermis than in the case of complex substances that act mainly on the skin surface. Active ingredients contained in coffee ferments have a nourishing and soothing effect, and due to the content of hydroxyl groups in their molecules, they are valuable moisturizing substances (humectants). These substances guarantee the high ability of ferments to provide a long-lasting and strong skin hydration and TEWL reduction. Due to their ability to reduce TEWL, they can also be a component of cosmetic or pharmaceutical products limiting the risk of irritations, e.g., after using cleansing cosmetics. Surfactants contained in their formulations show properties to increase TEWL and to disturb the hydrolipid barrier of the skin. Plant ferments can therefore be ingredients that decrease the ability of irritant factors that cause negative effects on the skin [52].

### 2.8. Determination of Sun Protection Factor (In Vitro)

As mentioned, the main advantage of plant ferments is the profile of the ingredients they contain. Due to the numerous data indicating the ability of green coffee to protect against UV radiation, the sun protection factor (SPF) value of green coffee ferments has been determined. The results are shown in Table 4.

According to literature data [53,54,55,56,57,58,59,60,61,62], the SPF of green coffee is from about SPF = 1–3 for extracts to about SPF = 3–4 for coffee seed oil. Green coffee products also have the ability to act as a booster for chemical UV filters, increasing their efficiency [62]. Green coffee ferments have slightly lower SPF values than green coffee extract. Fermentation time has a significant impact on their ability to protect against UV radiation. As the time of fermentation process increased, the SPF of green coffee ferments increased. The lowest SPF value was observed for the ferment after 7 days (SPF = 0.73), while the highest for the ferment obtained after 28 days of fermentation (SPF = 2.57). Plants are a source of substances considered to be natural sunscreens. The richest in this type of substance are cocoa and sea buckthorn fruits, walnuts, tea leaves, as well as marigold flowers and raspberry and strawberry seeds. The active ingredients of plants that give them the properties of natural sunscreens are mainly substances from a group of flavonoids, polyphenols, anthocyanins, as well as proteins and amino acids and vitamins. As indicated, coffee ferments are a rich source of these types of compounds that give them the ability to absorb UV radiation. The increase in the SPF value with the increase in fermentation time is most likely caused by the polymerization and depolymerization of the active substances contained in green coffee, which occur during the fermentation process [53].

## 3. Materials and Methods

### 3.1. Plant Material and Fermentation Procedure

The analyses were conducted using natural Arabica green coffee beans obtained from a local store. Beans were collected on controlled and ecological plantations. After purchase, the grains were ground using an electric mill. Kombucha starter cultures were purchased from a commercial source from Poland. Before starting the fermentation process, kombucha starter culture was stored under aseptic conditions in a refrigerator (4 °C) and consisted of acid broth and cellulose layer. Kombucha is composed primarily of acetic acid bacteria and osmophilic yeast. The most common bacteria are *Acetobacteraceae*, *Gluconobacter* and *Komagataeibacter* (*Komagataeibacter xylinus*, *Komagataeibacter interactus*, *Komagataeibacter rhaeticus*, *Komagataeibacter saccharivorans* and *Komagataeibacter kombuchae*). The yeast species included in this symbiotic consortium are those of the genera *Zygosaccharomyces*, *Candida*, *Torulaspora*, *Pichia, Brettanomyces*, *Schizosaccharomyces* and *Saccharomyces*. Initially, an infusion of green coffee was prepared in a sterile beaker by mixing 15 g of ground green coffee with humidity of 8% (3% *w*/*w*), 50 g of sucrose (final concentration 10.0% *m*/*v*) and 500 mL of hot distilled water (95 °C). The mixture prepared in this way was stirred every few minutes with a glass rod until the solution cooled down (about 25 °C, cooling bath, cooling time 30–40 min). The resulting green coffee decoction was then filtered twice through membrane filters into sterile glass beakers (1000 mL, 18 cm height and 8 cm diameter). Then, kombucha (3 g) was added to the filtrate and fermentation was carried out for 7, 14 and 28 days (in separate beakers) at room temperature (about 25 °C). After fermentation, the obtained kombucha was filtered and evaporated under reduced pressure at 40 °C. Ferments obtained after 7 days were signed as F7, after 14 days as F14 and after 28 days as F28. Green coffee decoction without kombucha was marked as GC.

### 3.2. Determination of Biologically Active Compounds

#### 3.2.1. The Determination of the Total Phenolic Content (TPC)

The concentration of total phenolic compounds in obtained kombucha ferments (F7, F14, F28) and green coffee beans’ extract was determined spectrophotometrically using the Folin–Ciocalteu method described by Singleton et al. [58] with some modifications. Gallic acid (GA) was used as standard. Briefly, 300 μL of aqueous solutions of dry ferments or green coffee beans extract at various concentrations was mixed with 1500 μL of Folin–Ciocalteu reagent (diluted 1:10). After 6 min of incubation, 1200 μL of a 7.5% sodium carbonate solution was added to the analyzed samples. Samples were mixed and incubated in the dark at room temperature (about 22 °C) for 2 h. Absorbance of the samples was read at λ = 740 nm on an AquamateHelion spectrophotometer (Thermo Fisher Scientific, Waltham, MA, USA). To calculate the total concentration of phenols in F7, F14, F28 and GC, a gallic acid (GA) calibration curve (in the 10–100 mg/mL concentration range) was used. The measurements were made in triplicate and the results obtained were averaged. The TPC results are presented as mg of GA equivalents (GAE) per g of dry weight.

#### 3.2.2. The Determination of the Total Flavonoids Content (TFC)

The concentration of flavonoids in the analyzed samples (F7, F14, F28 and GC) was assessed spectrophotometrically using aluminum nitrate nonahydrate. For this purpose, the method described by Matejić et al. [59] with modifications was used. Initially, 2400 μL of the previously prepared reaction mixture consisting of 80% C_2_H_5_OH, 10% Al(NO_3_)_3_ × 9H_2_O and 1M C_2_H_3_KO_2_ were mixed with 600 μL of the tested samples at various concentrations. After 40 min incubation at room temperature (about 22 °C) in the dark, the absorbance of the prepared mixtures was measured. Measurements were made at a wavelength λ = 415 nm using a FilterMax F5 AquamateHelion spectrophotometer (Thermo Fisher Scientific, Waltham, MA, USA). The total flavonoid concentration in the samples was calculated from the calibration curve for quercetin (Qu) hydrate (in the concentration range of 10–100 mg/mL). Measurements were made in triplicate for each sample. The TFC results are presented as mg of Q equivalents (QE) per g of dry weight.

#### 3.2.3. Determination of Bioactive Compounds by HPLC–UV-ESI–MS

The polyphenolic content was additionally quantified by high-performance liquid chromatography (HPLC). Two alkaloids’ content characteristic for green coffee beans: caffeine and trigonelline, was also determined.

The obtained kombucha ferments (F7, F14, F28) and green coffee beans’ extracts were analyzed to determine their main bioactive compounds using a HPLC (DionexUltiMate 3000 RS, Thermo Fisher Scientific, Waltham, MA, USA), coupled to both an ultraviolet-visible detector (DAD) and mass spectrometer (4000 QTRAP, Sciex, Framingham, MA, USA), equipped with an electrospray ionization source (ESI) and a triple quadrupole-ion trap mass analyzer.

Chromatographic separation was achieved with a gradient reverse-phase system. A 100 × 4.6 mm chromatographic column Kinetex 3.5 µm XB-C18 100 Å with iso-butyl side chains and with trimethylchlorosilane (TMS) end-capping stationary phase employed similar composition guard column was purchased from Phenomenex and maintained at 30 °C.

A binary solvent system comprising 0.1% (*v*/*v*) aqueous formic acid as solvent A and methanol as solvent B was used under gradient mode during 40 min of the run time. The elution conditions applied were as follow: 0.0–30.0 min 25–100% B, 30.0–35.0 min 100% B, 35.0–35.1 min 100–25% B, 35.1–40.0 min 25% B. Caffeine and trigonelline were monitored and quantified at 272 nm, and chlorogenic acids and derivatives, at 325 nm. The flow rate of the mobile phase was 0.6 cm^3^/min and injection volume was 3–50 μL.

Mass spectral (MS) data were collected in the negative and positive ionization mode with an electrospray source. The MS parameters were set as follows: curtain gas at 20 psi and nebulizer gas at 10 psi. Nitrogen was used as curtain and collision gas. Negative ionization mode source voltage −4500 V was applied for determination of phenolic compounds and positive ionization mode source voltage 5000 V for caffeine and trigonelline detection. Nitrogen was used as curtain and collision gas. The mass spectra were acquired with scans ranging from *m*/*z* 50 to *m*/*z* 1000. MS^2^ fragmentation was used to confirm the structure of detected compounds.

Chromatographic and spectral data were processed by using the instrument software Chromeleon 6.80 (Thermo Fisher Scientific, Waltham, MA, USA) and Analyst 1.5.1 (Sciex, Framingham, MA, USA), respectively.

Analytical standards of caffeoylquinic acids (CQA, two isomers: 3- and 5-CQA), caffeine and trigonelline were purchased from Sigma-Aldrich, Fluka, Chromadex. All standards used were of analytical grade (≥99% purity).

The identification of CQA isomers (3- and 5-CQA), caffeine and trigonelline was carried out by comparing the retention times obtained during the chromatographic separation of analytes in the GC, F7, F14 and F28 with the retention times of the individual compounds in respective standards and characterized by UV and mass spectra. Quantification of the above-mentioned substances were calculated on the basis of calibration curves, i.e., by using the external standard method. Quantification was performed using five-point analytical curves prepared in triplicate for caffeine (14.0–69.9 mg/L), trigonelline (8.2–41.2 mg/L), 3-CQA (1.2–12.3 mg/L) and 5-CQA (11.4–113.8 mg/L). For calculation of 4-CQA content, a 3-CQA calibration curve was used. The results were expressed in mg/100 g of dry weight of green coffee beans. The other chlorogenic acids and polyphenolic compounds were identified based on the elution order and MS spectra by molecular mass and ion fragmentation of each individual compound.

The identities of 14 compounds were determined.

From above, dicaffeoylquinic (diCQA) and feruloylquinic (FQA) acids as major compounds next to CQAs, were chosen for showing the phenolic profile more precisely. For that reason, quantification of diCQAs and FQAs was performed using the area of 5-CQA standard and molar extinction coefficients according to Reference [22], using the below equation:(1)$$c=\;\frac{RF\;\times \;\in_{1}\times MR_{2}\times \;A}{\in_{2}\times MR_{1}}$$
where:*c* is the concentration of interesting isomer in g/L,*RF* is the response factor of the 5-CQA standard in gmAV^−1^ min^−1^,∈_1_ is the molar extinction coefficient of 5-CQA in M^−1^ cm^−1^,∈_2_ is the molar extinction coefficient of interesting analogue or positional isomer in M^−1^ cm^−1^,*MR*_1_ is a molecular weight of 5-CQA in g/mol,*MR*_2_ is a relative molecular weight of interesting isomer in g/mol,*A* is an area of peak of the interesting isomer in mAV min.

Molar extinction coefficients (×10^4^) were as shown in Table 5.

### 3.3. Assessment of Antioxidant Activity

#### 3.3.1. DPPH Radical Scavenging Assay

The ability of the obtained kombucha and green coffee extract to scavenge free radicals was determined using the 1,1-diphenyl-2-picrylhydrazyl (DPPH) radical. The method described by Brand-Williams et al. [47] has been used. Initially, 33 μL of aqueous solutions of F7, F14 and F28 dry ferments and dry GC extract at concentrations of 100 µg/mL were mixed with 167 μL methanol solution of DPPH (4 mM) and transferred to a 96-well plate. The analyzed samples were thoroughly mixed by shaking. In the next step, the absorbance of the samples at 517 nm was measured. Measurements were made every 5 min for 30 min on a UV-VIS Filter Max λ = 5 spectrophotometer (Thermo Fisher Scientific, Waltham, MA, USA). Three independent replicates were performed for each concentration (for both kombucha and green coffee extract). Water with a DPPH solution was used as a control. The antioxidant capacity was expressed as a percentage of DPPH inhibition using the equation:(2)$$\%\;DPPH\;scavenging=\frac{Abs\;control\;-\;Abs\;sample}{Abs\;control}\;\times \;100$$
where:*Abs* control is the absorbance of the control sample (containing DPPH and water),*Abs* sample is the absorbance of the test sample (containing DPPH and test sample).

#### 3.3.2. Detection of Intracellular Levels of Reactive Oxygen Species (ROS)

In the next step, the intracellular level of free radicals was evaluated in the tested cell lines treated with the analyzed samples. In order to determine the ability of the analyzed kombucha and green coffee extract to generate or inhibit the intracellular production of ROS in HaCaT and BJ cells, a fluorogenic H_2_DCFDA dye was applied. After passive diffusion of H_2_DCFDA into the cells, it was deacetylated to a non-fluorescent compound. In the presence of ROS, it was oxidized and transformed into fluorescent 2′,7′-dichlorofluorescein (DCF).

To determine the intracellular level of ROS in HaCaTs and BJ cells, cells were seeded in 96-well plates at a density of 1 × 10^4^ cells per well. In the next step, cells were maintained in an incubator for 24 h. After that, DMEM was removed and replaced with 10 μM H_2_DCFDA (Sigma Aldrich, Saint Louis, MO, USA) dissolved in serum free DMEM medium. HaCAT and BJ cells were incubated in H_2_DCFDA for 45 min. Subsequently, the cells were incubated with various concentrations of kombucha and green coffee bean extract. Cells treated with 1 mM hydrogen peroxide (H_2_O_2_) were used as a positive control. DCF fluorescence was measured every 30 min for 90 min using a FilterMax F5 microplate reader (Thermo Fisher Scientific, Waltham, MA, USA). Measurements were made at a maximum excitation of λ = 485 nm and emission spectra of λ = 530 nm. All tested samples were made in 4 replicates.

#### 3.3.3. Determination of Superoxide Dismutase (SOD) Activity

To determine the effect of kombucha and green coffee extract on the activity of the enzyme that plays an important role in defense against reactive oxygen species, a kit for assessing the activity of superoxide dismutase (ab65354, Abcam, Cambrige, UK) was used. F7, F14, F28 and GC in concentrations of 100, 500 and 1000 µg/mL were used for the analysis. Recombinant human SOD 1 protein (ab112193, Abcam, Cambrige, UK) was used to prepare the standard curve. Samples were prepared in 96-well plates (clear bottoms) and the analysis was performed according to the manufacturer’s instructions. Briefly, 200 μL of 2-(4-iodophenyl)-3-(4-nitrophenyl)-5-(2,4-disulfo-phenyl)-2H-tetrazolium, monosodium salt (WST) working solution was added to each well. Then, test samples were prepared by adding 20 μL Enzyme Working Solution (EWS) and 20 μL of analyzed samples (with a final concentration of 100, 250 and 1000 µg/mL) to these wells. Three different blank samples were also prepared as recommended. Blank 1 was prepared by adding 20 μL EWS and 20 μL ddH_2_O to the wells. To blank 2, 20 μL Dilution Buffer (DB) and 20 μL F7, F14, F28 or GC (with a final concentration of 100, 250 and 1000 µg/mL) was added. 20 μL of DB and 20 μL ddH_2_O were added to blank 3. All prepared samples were mixed thoroughly by shaking and incubated at 37 °C for 20 min. Subsequently, the absorbance of the analyzed samples was measured at λ = 450 nm using a microplate reader (FilterMax F5, Thermo Fisher, Waltham, MA, USA). All samples were prepared in duplicate according to the manufacturer’s instructions. The ability to inhibit SOD activity was calculated from the equation:(3)$$\%\;SOD\;Activity=\frac{\left(Ablank1-\;Ablank3\right)\;-\;\left(Asample\;-\;Ablank2\right)}{\left(Ablank1\;-\;Ablank3\right)}\;\times \;100$$

### 3.4. Cell Culture

As part of this work, two cell lines located in different layers of the skin were used. HaCaT cells (normal human keratinocytes) were obtained from CLS Cell Lines Service (CLS Cell Lines Service GmbH, Eppelheim, Germany). Fibroblasts (BJ) cells (ATCC^®^CRL-2522 ™, ATCC, Manassas, VA, USA) were purchased from the American Type Culture Collection (Manassas, VA 20108, MA, USA). Both cell lines were grown in DMEM (Dulbecco’s Modification of Eagle’s Medium, Biological Industries) with L-glutamine, 4.5 g/L glucose and sodium pyruvate. The medium was supplemented with 10% FBS (Fetal Bovine Serum, Gibco, Waltham, MA, USA) and 1% antibiotics (100 U/mL penicillin and 1000 µg/mL streptomycin, Gibco). Cells were maintained in an incubator at 37 °C in a humid atmosphere of 95% air and 5% carbon dioxide (CO_2_).

### 3.5. Cell Viability Assay

To perform cytotoxicity tests on the tested cell lines, the medium was aspirated, and the cells attached to the bottom were washed twice with sterile PBS (phosphate buffered saline, Gibco). Then, the cell layer was trypsinized using Trypsin/tetraacetylethylenediamne (EDTA) (Gibco) and the detached cells were centrifuged and resuspended in fresh DMEM medium. In the next step, the cells were plated in 96-well plates. After the cells were attached to the bottom, they were incubated with various concentrations (10, 100, 250, 500 and 1000 μg/mL) of kombucha ferments and green coffee bean extract. Then, the cells were inserted into the incubator and incubated for 24 h with the analyzed samples. The control was cells (separately HaCaT and BJ) cultured in DMEM without the addition of extracts.

#### 3.5.1. Neutral Red Uptake Assay

The neutral red uptake test (Sigma Aldrich) was applied in this work to evaluate the viability of skin cells treated with the analyzed samples (F7, F14, F28 and GC). The tests were carried out in accordance with the procedure previously described by us based on the methodology proposed by Borenfreund and Puerner [60]. The average optical density of the control cells (not incubated with analyzed samples) was set to 100% viability and was used to calculate the percentage of viable cells in the experimental samples. The experiments were repeated three times using four wells for each analyzed concentration.

#### 3.5.2. Alamar Blue Assay

To assess the cytotoxicity of the tested samples and evaluate their effect on cell viability, the Alamar Blue (AB) test (Sigma, R7017) was applied. After 24 h exposure of HaCaT cells and fibroblasts to F7, F14, F28 and GC in a concentration range of 10–1000 μg/mL, a solution of resazurin with a final concentration of 60 μM was added to the wells. The plates were then incubated for 2 h at 37 °C in the dark in an incubator. After incubation, fluorescence of individual wells was measured at λ = 570 nm using a microplate reader (FilterMax F5, Thermo Fisher Scientific, Waltham, MA, USA). Three independent experiments were performed to assess cytotoxicity using the AB assay. Results are expressed in graphs as a percentage of cell viability compared to controls (100%).

#### 3.5.3. Lactate Dehydrogenase (LDH) Cytotoxicity Assay

The cytotoxicity of kombucha (F7, F14 and F28) and green coffee extract were also assessed using a kit from G-Biosciences (LDH Cytoscan ™ Cytotoxicity Test). The principle of this assay was based on the conversion of lactate to pyruvate in the presence of lactate dehydrogenase. The test was carried out according to the manufacturer’s instructions. Analysis was performed on 96-well plates with seeded HaCaT and BJ cells in DMEM medium. After attachment to the bottom, the cells were treated with kombucha and green coffee bean extract in the concentration range of 10–1000 μg/mL. To prepare Spontaneous LDH Activity Control of LDH activity, no compound was added to the wells. After incubation with tested compounds, the medium was removed and then the culture supernatant was collected and incubated with 50 μL of reaction mixture for 30 min at 25 °C. The reaction was then stopped by adding 50 μL Stop Solution. To determine lactate dehydrogenase activity, absorbance at λ = 490 nm and λ = 680 nm was measured using a FilterMax F5 microplate reader (Thermo Fisher Scientific, Waltham, MA, USA). Cytotoxicity of the analyzed samples was calculated using the equation:(4)$$\%\;Cytotoxicity=\frac{Compound\;Treated-Spontaneous\;LDH\;Activity}{Maximum\;LDH\;release-Spontaneous\;LDH\;Activity}\;\times \;100\;$$

### 3.6. Assessment of Matrix Metallopeptidases Inhibition

#### 3.6.1. Determination of Anti-Collagenase Activity

To assess the ability of the obtained kombucha ferments (F7, F14 and F28) and the green coffee bean extract to inhibit collagenase activity, a fluorometric kit (Abcam, ab211108) was applied. Analysis was carried out for all kinds of kombucha ferments and green coffee extract in concentrations of 100, 500 and 1000 µg/mL. Analysis was performed on a 96-well plate with a transparent flat bottom. Initially, collagenase (COL) was dissolved in a collagenase analysis buffer (CAB). Test samples were prepared by adding the analyzed samples to COL and CAB. Inhibitor control samples were prepared by mixing the collagenase inhibitor (1,10-phenanthroline (80 mM)) with collagenase and CAB buffer. Enzyme control wells were prepared by mixing diluted COL with CAB. The CAB buffer was used as a background control. The prepared samples were incubated for 15 min at room temperature. In addition, a reaction mixture was prepared by mixing the collagenase substrate with CAB. The reaction mixture prepared in this way was added to all analyzed samples and mixed thoroughly. In the next step, fluorescence was measured at excitation wavelength λ = 490 nm and emission λ = 520 nm using a microplate reader (FilterMax F5, Thermo Fisher Scientific, Waltham, MA, USA). The measurement was made in kinetic mode for 60 min at 37 °C. All samples were prepared in duplicate according to the manufacturer’s instructions. The ability to inhibit COL activity by F7, F14, F28 and GC was calculated by the equation:(5)$$\%\;relative\;COL\;inhibition=\frac{enzyme\;control-sample}{enzyme\;control}\;\times \;100$$

#### 3.6.2. Determination of Anti-Elastase Activity

To determine the possibility of inhibiting another matrix metalloproteinase, neutrophil elastase (NE), a fluorometric kit (Abcam, ab118971) was applied. The analysis used analogous concentrations of the tested samples as in the case of the test described above evaluating the possibility of collagenase inhibition. Analysis was performed in 96-well black plates (transparent bottoms). The test procedure was based on the instructions provided by the manufacturer. Initially, NE enzyme solutions, NE substrate and inhibitor control (SPCK) were prepared according to the instructions. Then, diluted NE solution was added to all wells. Test samples, inhibitor control and enzyme control (Assay Buffer) were added to subsequent wells. All samples were prepared in duplicate. After all reagents were added, the samples were mixed. The plate was then incubated at 37 °C for 5 min. In the meantime, a reaction mixture was prepared by mixing the Assay Buffer and NE substrate. The mixture was added to each well and mixed thoroughly. Fluorescence was measured immediately at excitation wavelength λ = 400 nm and emission λ = 505 nm using a microplate reader (FilterMax F5, Thermo Fisher Scientific, Waltham, MA, USA). The kinetic mode was used (30 min at 37 °C). The ability to inhibit NE activity by the analyzed samples (F7, F14, F28 and GC) was calculated from the equation:(6)$$\%\;relative\;NE\;activity=\frac{\Delta RFU\;test\;inhibitor}{\Delta \;RFU\;Enzyme\;control}\;\times \;100$$

### 3.7. Transepidermal Water Loss (TEWL) and Skin Hydration Measurements

TEWL and skin hydration measurements were conducted using TEWAmeter TM 300 probe and Corneometer CM 825 probe connected to a MPA adapter (Courage + Khazaka Electronic, Köln, Germany). The study was conducted on 15 volunteers. Five areas (2 × 2 cm in size) were marked on the forearm skin. 0.2 mL of 100 µg/mL solution of F7, F14, F28 and GC (aqueous solutions of dry ferments and extract) was applied to 5 fields. One field (control field) was not treated with any sample. Sample solutions were gently spread over every skin fragment, and then rinsed with distilled water and dried with a paper towel. After 60, 180 and 360 min, the hydration and TEWL measurements were taken. The final result was the arithmetic mean (from each volunteer) of 5 independent measurements (skin hydration) and 20 measurements (TEWL).

### 3.8. Determination of Sun Protection Factor (In Vitro)

Sun protection factor (SPF) of green coffee extract and fermented green coffee was determined according to the method described by the Mansur Equation. SPF was determined by measuring the absorbance of aqueous solution (50 µg/mL) of the dried extract or the ferment within the wavelength range from 290 to 320 nm at 5 nm intervals. SPF was calculated from the Mansur Equation [61]:(7)$$\mathrm{SPF}\;=\;\mathrm{CF}\;\times \;\sum_{290}^{320}\;[\;\mathrm{EE}\;\left(\lambda \right)\times I\left(\lambda \right)\times \mathrm{ABS}\;\left(\lambda \right)\;]$$
where:EE (λ)—erythemal effect spectrum,I (λ)—solar intensity spectrum,Abs (λ)—absorbance of sunscreen product,CF—correction factor (= 10),E (λ) × I(λ)—values determined by Sayre were used [62].

### 3.9. Statistical Analysis

Values of different parameters were expressed as the mean ± standard deviation (SD). The two-way analysis of variance (ANOVA) and Bonferroni post-test between groups were performed at the *p*-value level of <0.05 to evaluate the significance of differences between values. Statistical analysis was performed using GraphPad Prism 8.4.3 (GraphPad Software, Inc., San Diego, CA, USA).

## 4. Conclusions

The results of the performed analysis show that the investigated green coffee extracts and ferments with kombucha are characterized by a significant content of biologically active compounds, which values correlate with their antioxidant potential. Green coffee extracts and ferments subjected to a 28-day fermentation process were characterized by the highest antioxidant capacity. Coffee extracts and ferments obtained after 14 days of fermentation were characterized by a high ability to inhibit collagenase and elastase. In addition, these ferments can have a positive effect on skin hydration and reduce transepidermal water loss. The analyzed samples showed no cytotoxicity to skin cells, both keratinocytes and fibroblasts.

The conducted research shows that both the analyzed green coffee bean extracts and the ferments obtained with kombucha may be a valuable source of bioactive substances and can be used in cosmetic and dermatological products.

## Figures and Tables

**Figure 1 molecules-25-05394-f001:**
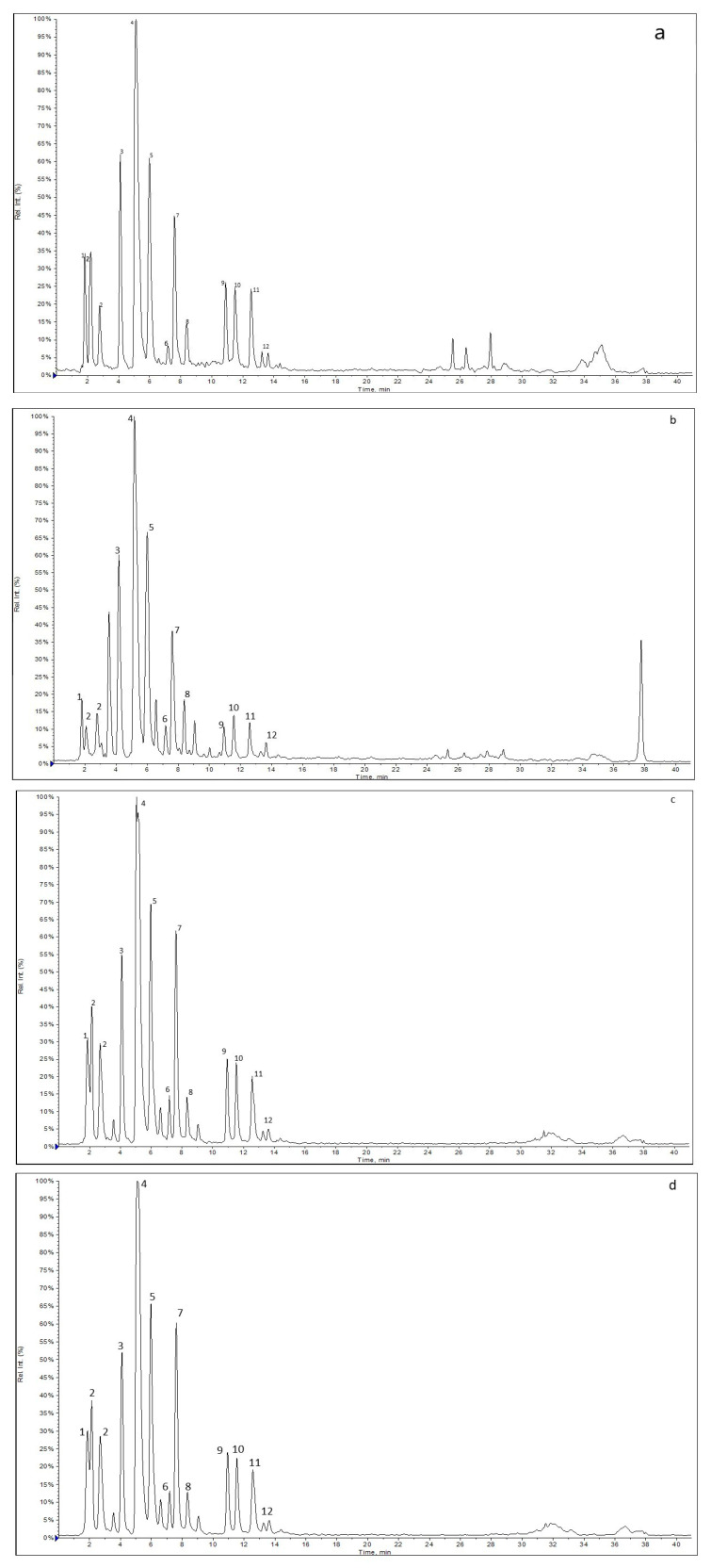
Extracted ion chromatograms (XIC) obtained for (**a**) GC, (**b**) F7, (**c**) F14 and (**d**) F28 extracts.

**Figure 2 molecules-25-05394-f002:**
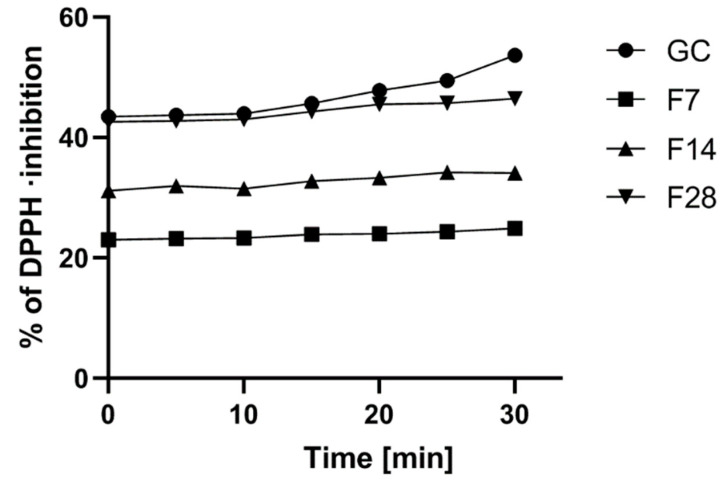
Kinetics of the absorbance changes in DPPH solutions of green coffee beans’ extract and kombucha ferments. Values are mean of three replicate determinations (*n* = 3).

**Figure 3 molecules-25-05394-f003:**
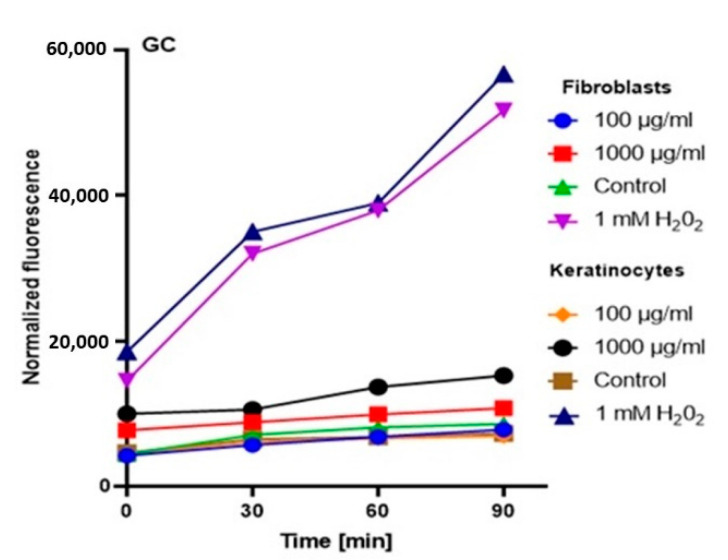
The effect of Green coffee extract on the DCF fluorescence in fibroblasts and HaCaT cells. The data are expressed as the mean  ±  SD of 3 independent experiments, each of which consisted of 3 replicates per treatment group.

**Figure 4 molecules-25-05394-f004:**
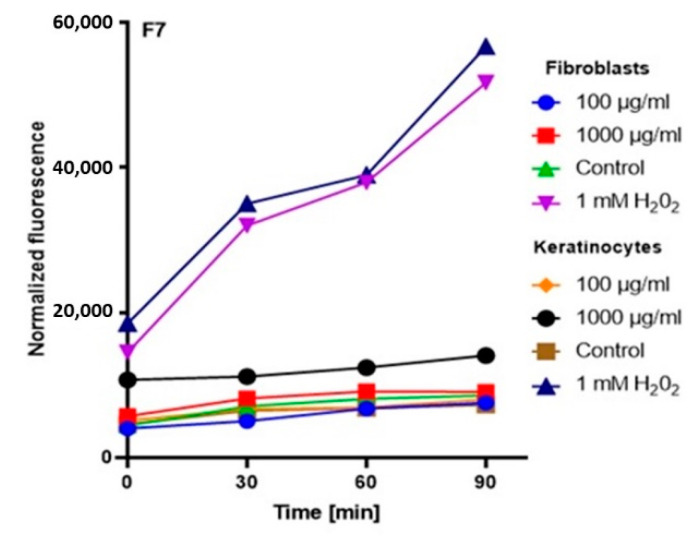
The effect of kombucha ferments’ extracts (F7) on the DCF fluorescence in fibroblasts and HaCaT cells. The data are expressed as the mean  ±  SD of 3 independent experiments, each of which consisted of 3 replicates per treatment group.

**Figure 5 molecules-25-05394-f005:**
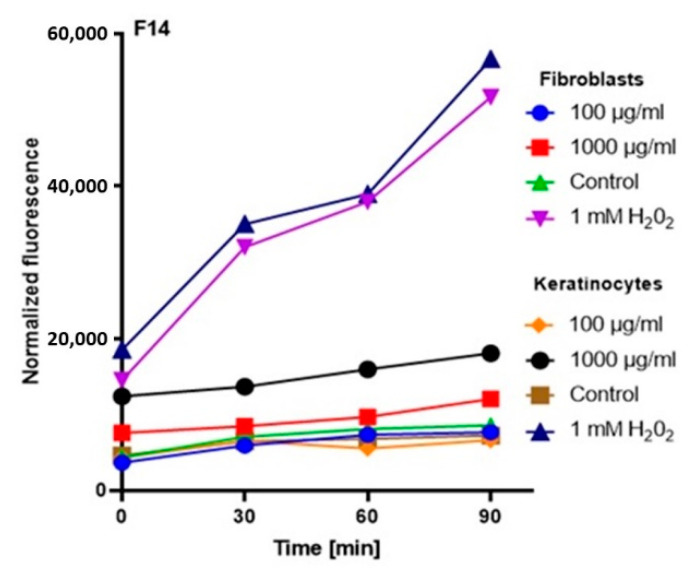
The effect of kombucha ferments’ extracts (F14) on the DCF fluorescence in fibroblasts and HaCaT cells. The data are expressed as the mean  ±  SD of 3 independent experiments, each of which consisted of 3 replicates per treatment group.

**Figure 6 molecules-25-05394-f006:**
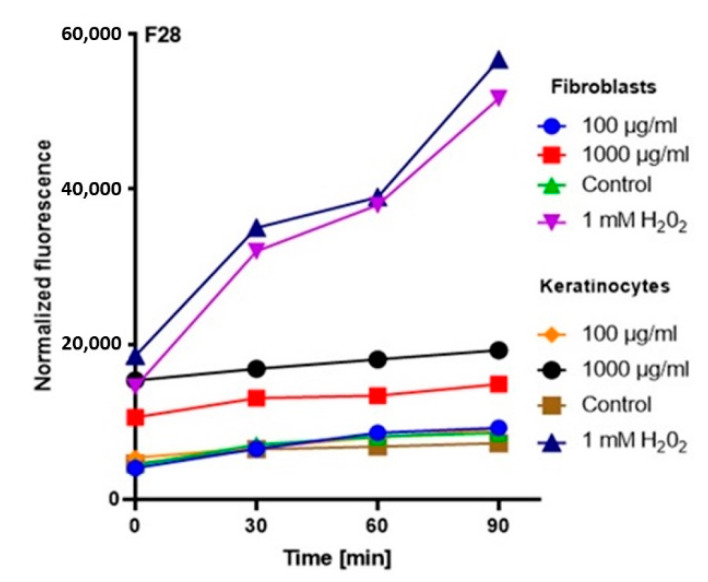
The effect of kombucha ferments’ extracts (F28) on the DCF fluorescence in fibroblasts and HaCaT cells. The data are expressed as the mean  ±  SD of 3 independent experiments, each of which consisted of 3 replicates per treatment group.

**Figure 7 molecules-25-05394-f007:**
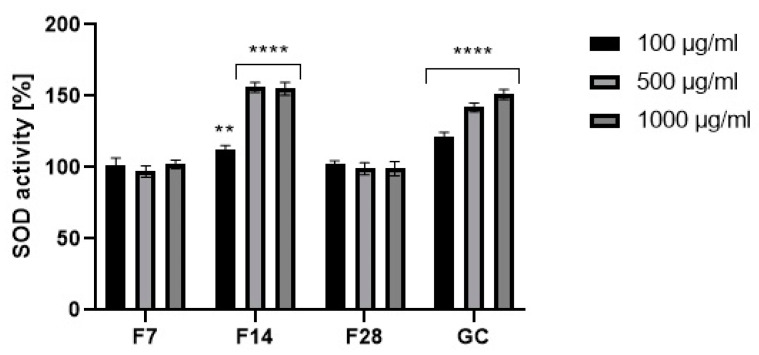
Effect of coffee beans’ extract and kombucha ferments on superoxide dismutase activity. Data are the mean ± SD of three independent experiments, in which each concentration was tested in duplicate. **** *p* < 0.0001, ** *p* < 0.002.

**Figure 8 molecules-25-05394-f008:**
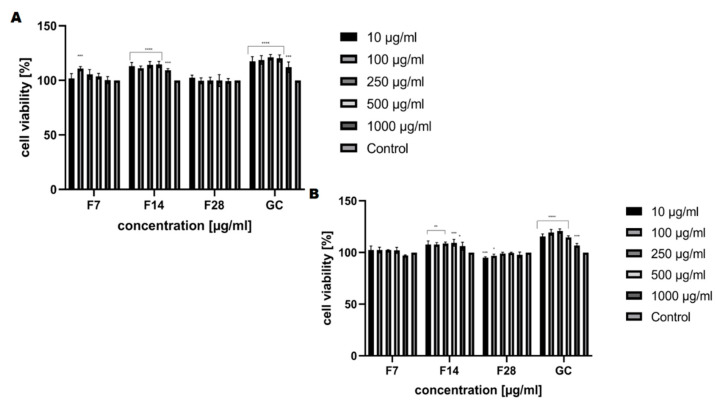
The reduction of resazurin after 24 h exposure to the ferments obtained from green coffee beans (1–1000 μg/mL) in cultured (**A**) fibroblasts (BJ) and (**B**) keratinocytes (HaCaT). Data are the mean ± SD of three independent experiments, each of which consists of three replicates per treatment group. For BJ, **** *p* < 0.0001, *** *p* < 0.0007 versus the control (100%). For HaCaT, **** *p* < 0.0001 *** *p* < 0.0008, ** *p* < 0.003, * *p* < 0.04 versus the control (100%).

**Figure 9 molecules-25-05394-f009:**
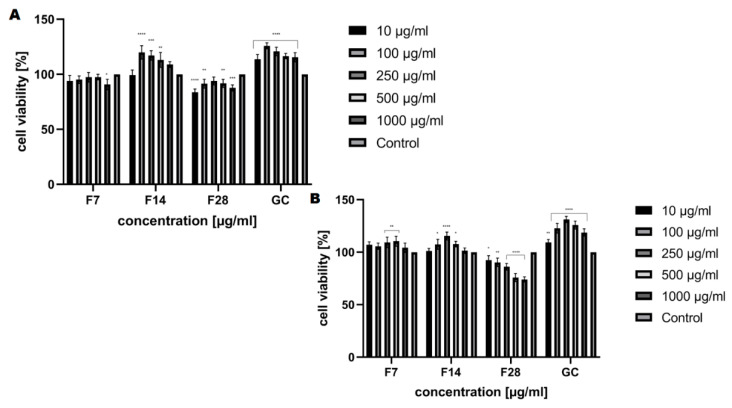
The effect of increasing concentrations of the ferments obtained from green coffee beans (1–1000 μg/mL) on Neutral Red Dye uptake in cultured (**A**) fibroblasts (BJ) and (**B**) keratinocytes (HaCaT) after 24 h of exposure. Data are the mean ± SD of three independent experiments, each of which consists of four replicates per treatment group. For BJ, **** *p* < 0.0001, *** *p* < 0.0003, ** *p* < 0.03, * *p* < 0.01 versus the control (100%). For HaCaT, **** *p* < 0.0001 ** *p* < 0.009, * *p* < 0.02 versus the control (100%).

**Figure 10 molecules-25-05394-f010:**
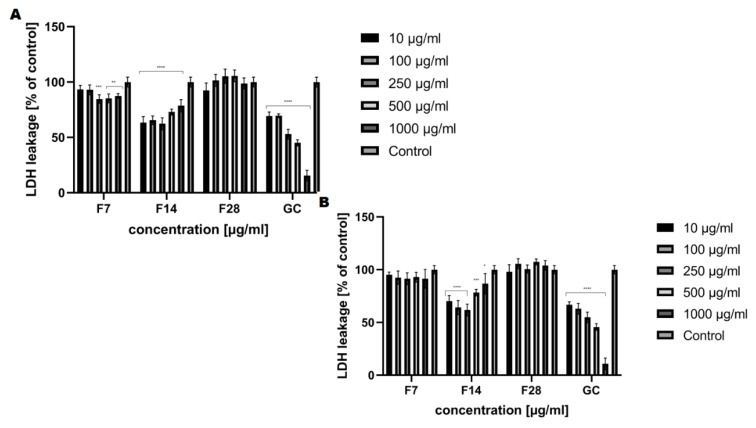
The effect of increasing concentrations of the ferments obtained from green coffee beans (1–1000 μg/mL) on LDH uptake in cultured (**A**) fibroblasts (BJ) and (**B**) keratinocytes (HaCaT) after 24 h of exposure. Data are the mean ± SD of three independent experiments, each of which consists of four replicates per treatment group. For BJ, **** *p* < 0.0001, *** *p* < 0.0003, ** *p* < 0.03 versus the control (100%). For HaCaT, **** *p* < 0.0001, *** *p* < 0.009, * *p* < 0.02 versus the control (100%).

**Figure 11 molecules-25-05394-f011:**
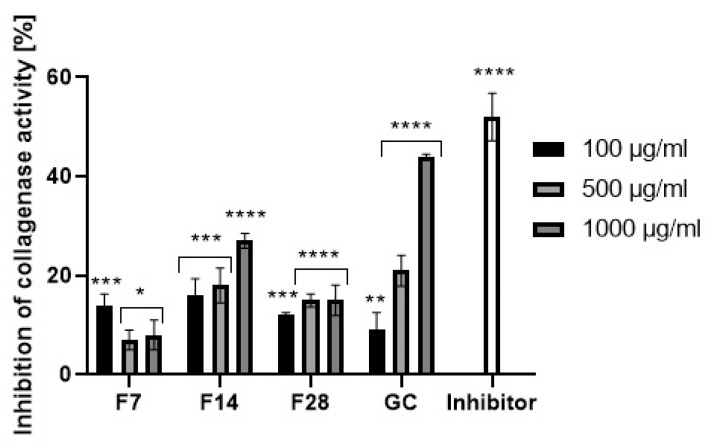
Collagenase inhibitory activity of coffee beans and kombucha ferments. Data are the mean of three independent experiments, each consisting of two replicates per treatment group. **** *p* < 0.0001, *** *p* < 0.0002, ** *p* < 0.02, * *p* < 0.01.

**Figure 12 molecules-25-05394-f012:**
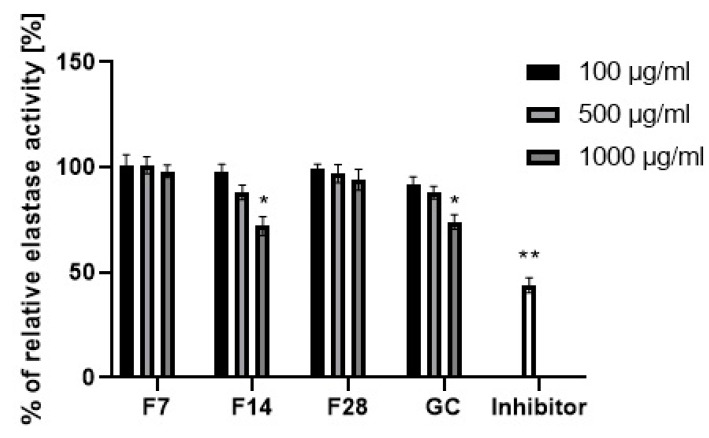
Elastase activity of coffee beans and kombucha ferments. Data are the mean of three independent experiments, each consisting of two replicates per treatment group. ** *p* < 0.03, * *p* < 0.01.

**Figure 13 molecules-25-05394-f013:**
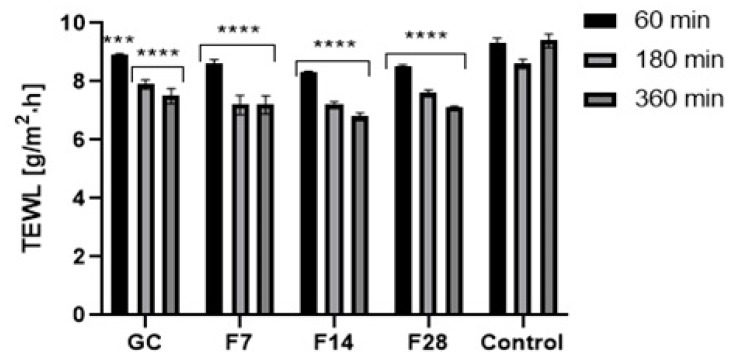
Influence of coffee beans’ and kombucha ferments’ extracts on transepidermal water loss (TEWL). Data are the mean ± SD of five independent measurements. *** *p* < 0.0003, **** *p* < 0.0001.

**Figure 14 molecules-25-05394-f014:**
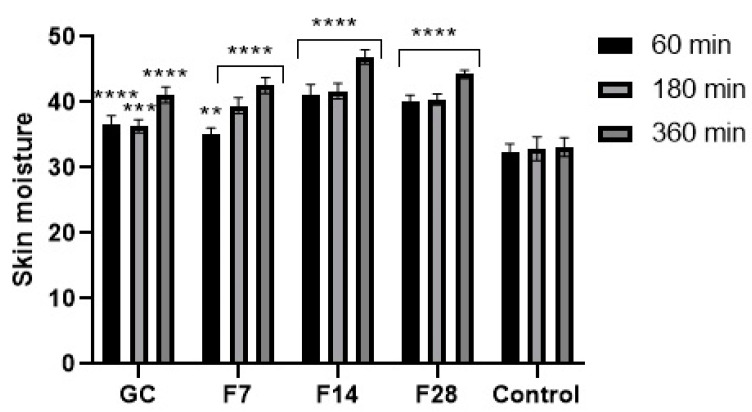
Influence of coffee beans and kombucha ferments on skin hydration. Data are the mean ± SD of five independent measurements. **** *p* < 0.0001, *** *p* < 0.0003, ** *p* < 0.03.

**Table 1 molecules-25-05394-t001:** Total phenolic (TPC) and flavonoids (TFC) content in green coffee extract and green coffee ferments (DW—dry weight of ferments or extract, GAE—gallic acid, QE—quercetin).

	TPC (mg GAE/g DW)	TFC (mg QE/g DW)
GC	630.05 ± 5.20 ^a^	156.84 ± 4.11 ^a^
F7	106.76 ± 3.72 ^b^	17.02 ± 2.33 ^b^
F14	243.14 ± 3.22 ^c^	51.10 ± 2.84 ^c^
F28	392.56 ± 1.22 ^d^	66.53 ± 3.84 ^d^

^a,b,c,d^: Different letters on the charts indicate significant differences between groups (*p* < 0.05).

**Table 2 molecules-25-05394-t002:** Polyphenols detected using HPLC-UV-ESI-MS.

No.	Retention Time (min)	Molecular Formula	Molar Mass (Da)	Precursor Ion [M–H]^−^ *m*/*z*	Main Productions MS^2^ *m*/*z*	Identification
**Negative-Ion Mode**						
1	1.8	C_9_H_8_O_4_	180.2	179 [M–H]^−^	135 [M-COOH]^−^, 107 [M-C_3_H_5_O_2_]^−^, 71 [M-C_6_H_5_O_2_]^−^, 59 [M-C_7_H_5_O_2_]^−^	Caffeic acid
2	2.0/2.8	C_7_H_12_O_6_	192.2	191 [M–H]^−^	127 [M-H-H_2_O-HCOOH]^−^, 85 [M-C_3_H_7_O_4_]^−^, 59 [M-C_5_H_9_O_4_]^−^	Quinic acid
3	4.1	C_16_H_18_O_9_	354.3	353 [M–H]^−^	191 [M-3H_2_O-C_6_H_5_O_2_]^−^, 179 [M-3H_2_O-C_6_H_4_-COOH]^−^,135 [M-3H_2_O-C_6_H_4_-C_2_HO_4_]^−^	3-Caffeoylquinic acid
4	5.2	C_16_H_18_O_9_	354.3	353 [M–H]^−^	191 [M-3H_2_O-C_6_H_5_O_2_]^−^, 179 [M-3H_2_O-C_6_H_4_-COOH]^−^	5-Caffeoylquinic acid
5	6.0	C_16_H_18_O_9_	354.3	353 [M–H]^−^	173 [M-C_6_H_3_-2OH-C_2_H_2_-COOH]^−^, 191 [M-3H_2_O-C_6_H_5_O_2_]^−^, 179 [M-3H_2_O-C_6_H_4_-COOH]^−^, 135 [M-3H_2_O-C_6_H_4_-C_2_HO_4_]^−^	4-Caffeoylquinic acid
6	7.2	C_16_H_18_O_8_	338.3	337 [M–H]^−^	191 [M-C_9_H_7_O_11_]^−^, 173 [M-C_9_H_9_O_3_]^−^, 163 [M-C_7_H_11_O_5_]^−^	5-p-Coumaroylquinic acid
7	7.6	C_17_H_20_O_9_	368.3	367 [M–H]^−^	191 [M-C_10_H_9_O_3_]^−^, 133 [M-C_13_H_15_O_4_]^−^, 173 [M-C_10_H_11_O_4_]^−^	3-Feruloylquinic acid
8	8.4	C_17_H_20_O_9_	368.3	367 [M–H]^−^	191 [M-C_10_H_9_O_3_]^−^, 173 [M-C_10_H_11_O_4_]^−^	5-Feruloylquinic acid
9	10.9	C_25_H_24_O_12_	516.4	515 [M–H]^−^	353 [M-C_9_H_7_O_3_]^−^, 335 [M-C_9_H_9_O_4_]^−^, 179 [M-C_16_H_17_O_8_]^−^	3,4-Dicaffeoylquinic acid
10	11.5	C_25_H_24_O_12_	516.4	515 [M–H]^−^	353 [M-C_9_H_7_O_3_]^−^	3,5-Dicaffeoylquinic acid
11	12.5	C_25_H_24_O_12_	516.4	515 [M–H]^−^	353 [M-C_9_H_7_O_3_]^−^, 179 [M-C_16_H_17_O_8_]^−^	4,5-Dicaffeoylquinicacid
12	13.2	C_26_H_26_O_12_	530.5	529 [M–H]^−^	353 [M-C_10_H_9_O_3_]^−^, 367 [M-C_9_H_7_O_3_]^−^, 191 [M-C_16_H_19_O_8_]^−^	3-Caffeoyl,5-feruloylquinic acid
**Positive-Ion Mode**						
13	1.8	C_7_H_7_NO_2_	137.1	138 [M + H]^+^	92 [M-H-H-CO_2_]^+^, 94 [M-CO_2_]^+^, 78 [M-H- H-CO_2_-CH_3_]^+^	Trigonelline
14	5.2	C_8_H_10_N_4_O_2_	194.2	195 [M + H]^+^	138 [M-CO-N-CH_3_]^+^, 110 [M-CO-N-CH_3_-CO]^+^	Caffeine

**Table 3 molecules-25-05394-t003:** Quantification results obtained for GC, F7, F14 and F28 extracts. Values are means ± standard deviation (SD) of triplicate. LOD—limit of detection.

Compound	Content (mg/100g of Dry Weight of Green Coffee Beans)			
	GC	F7	F14	F28
5-CQA	4944.9 ± 229.9	2486.7 ± 32.3	3198.4 ± 15.3	2649.0 ± 34.7
4-CQA	562.9 ± 14.7	483.7 ± 3.8	426.2 ± 6.9	531.1 ± 5.5
3-CQA	397.9 ± 13.0	390.7 ± 13.0	275.5 ± 0.3	429.1 ± 1.8
3.5-diCQA	61.2 ± 0.7	49.9 ± 4.5	<LOD	71.2 ± 2.5
4.5-diCQA	101.2 ± 2.0	41.5 ± 3.6	<LOD	61.5 ± 2.3
3.4-diCQA	44.0 ± 0.6	33.7 ± 2.2	<LOD	50.1 ± 1.9
3-FQA	20.7 ± 0.5	17.7 ± 2.7	9.2 ± 2.4	19.0 ± 0.3
4-FQA	<LOD	<LOD	<LOD	<LOD
5-FQA	544.1 ± 1.3	345.8 ± 5.9	319.3 ± 4.6	350.4 ± 2.7
**Sum of quantified phenolic compounds**	**6677.0**	**3849.7**	**4228.6**	**4161.3**
Caffeine	1956.1 ± 29.0	1403.4 ± 1.2	1165.3 ± 4.1	1349.4 ± 1.0
Trigonelline	1205.5 ± 22.1	922.3 ± 1.2	910.4 ± 1.4	839.8 ± 1.0
**Sum of quantified compounds**	**9838.6**	**6175.5**	**6304.4**	**6350.6**

**Table 4 molecules-25-05394-t004:** Ability of green coffee to protect against UV radiation.

Sample	GC	F7	F14	F28
SPF	3.15 ± 0.22	0.73 ± 0.09	2.14 ± 0.12	2.57 ± 0.13

**Table 5 molecules-25-05394-t005:** Molar extinction coefficients.

Compound	Molar Extinction Coefficient (×10^4^) (M^−1^ cm^−1^)	λ_max_ (nm)
5-CQA	1.95	330
4-CQA	1.80	
3-CQA	1.84	
3,4-diCQA	3.18	
3,5-diCQA	3.16	
4,5-diCQA	3.32	
5-FQA	1.93	325
4-FQA	1.95	
3-FQA	1.90	

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
