# Peer review of "Effect of Fermentation Time on Antioxidant and Anti-Ageing Properties of Green Coffee Kombucha Ferments"

_molecules, 2020, doi:10.3390/molecules25225394_

Round 1

Reviewer 1 Report

Original work on a new substrate for Kombusha fermentation. Cosmetic application is interesting.

The results are clearly introduced and exposed.

Suggestions of improvments :

- in the introduction : page 2 : " currently attemps are being made to obtain bioferements using SCOBY" ==> bioferements or bioferments ?

- Improvments in the Fermentation Procedure part of Materials and methods have to be done : there is a lack of precision in certain point of the infusion and fermentation procedure

  • 50 g of sugar ==> we suppose sucrose, but you have to precise (works had ever been realized with glucose, mannose...)
  • about the 15g of ground green coffee : is it dry ? otherwise, you have to precise the % of humidity
  • time of infusion is vague : "until the solution cooled downed" ==> the infusion tank is opened/closed ? time and condition of infusion can impact the molecule transfer from coffee
  • what about the fermentation beakers geometry ==> For the tea kombucha fermentation, it's known that the geometry impacts the processus of fermentation
  • what is the quantity of kombusha inoculated ???

- questions :

  • did you realize replicates of the fermentation ? kombucha fermentations are known to have variability, and problem or reproductibility...
  • did you measure the sugar/ alcool/ acid kinetics ? in order to know if the fermentation is finished at F28 (all the sugar consumned)
  • page 18 : i'm not sure to well undersant... for calculation 4-CQA content .... the results were expressed in mg/100 g of dry weight ==> but dry weight of what ? coffee extracts ? Do you dry the samples at the end of fermentation or it's related to the quantity of coffee introduced for the infusion ?

Author Response

Dear Reviewer,

Thank you very much for your suggestions and for taking the time to do this review. I am sending our answers to your comments.

- in the introduction : page 2 : " currently attemps are being made to obtain bioferements using SCOBY" ==> bioferements or bioferments ?

Thank you very much for your suggestion. It was of course letter mistake.

- Improvments in the Fermentation Procedure part of Materials and methods have to be done : there is a lack of precision in certain point of the infusion and fermentation procedure

Thank you very much for each of these comments. The manuscript has been corrected with your suggestions and the fermentation procedure has been improved.

  • 50 g of sugar ==> we suppose sucrose, but you have to precise (works had ever been realized with glucose, mannose...)

It was of course sucrose. It was added to manuscript (p. 17).

  • about the 15g of ground green coffee : is it dry ? otherwise, you have to precise the % of humidity

We checked the humidity of coffee beans ground before their extraction and fermentation. It was 8%. The information was added to manuscript.

  • time of infusion is vague : "until the solution cooled downed" ==> the infusion tank is opened/closed ? time and condition of infusion can impact the molecule transfer from coffee

The extractor (infusion tank) was closed during extraction of green coffee beans. Obtained extracts was cooled to about 25oC in cooling bath in 30-40 minutes. The information was added to manuscript (p. 17, section: plant material and fermentation procedure).

  • what about the fermentation beakers geometry ==> For the tea kombucha fermentation, it's known that the geometry impacts the processus of fermentation

The Fermentation process was carried out in 1000 ml beakers (height: 18 cm, diameter: 8 cm).

  • what is the quantity of kombusha inoculated ???

The quantity of kombucha was 3 g for each of fermentations. The information was added to manuscript (p. 17, section: plant material and fermentation procedure).

- questions :

  • did you realize replicates of the fermentation ? kombucha fermentations are known to have variability, and problem or reproductibility...

Each of fermentation process was carried out in triplicate. The results presented in the manuscript are the average results obtained for three ferments.

  • did you measure the sugar/ alcool/ acid kinetics ? in order to know if the fermentation is finished at F28 (all the sugar consumned)

Dear reviewer,

kinetics has not been studied. In this work, we analyzed the effect of fermentation time on the properties of the obtained ferments. When we observed that the properties of the ferments was worse after 28 days, we finished the fermentation process at this point.

  • page 18 : i'm not sure to well undersant... for calculation 4-CQA content .... the results were expressed in mg/100 g of dry weight ==> but dry weight of what ? coffee extracts ? Do you dry the samples at the end of fermentation or it's related to the quantity of coffee introduced for the infusion ?

For the calculation of the 4-CQA content, and other phenolic compounds (HPLC analysis), the results were expressed in mg/100 g of dry weight of green coffee beans introduced into the infusion.

Results of other studies were calculated on the dry weight of ferments. After fermentation, we evaporated the filtrate under reduced pressure at 40 ° C. This information was added to the manuscript.

Authors

Reviewer 2 Report

This study investigated the “Effect of kombucha fermentation time of green coffee beans on properties of the obtained ferments” The topic is interesting, and manuscript is overall well-written. The authors concluded based on their experimental data. The strength of this paper is the providing many figures and tables.

Minor comments:

The title of this paper is somewhat confusing. Title should be restated with inclusion of keywords and the conclusion of this study.

Abstract is too long and hard to read all. Write briefly with including a brief background, purpose, methods, results, and conclusion.

What microorganisms were used for fermentation? Include in the M&M.

Author Response

Dear Reviewer,

Thank you very much for your suggestions and for taking the time to do this review. I am sending our answers to your comments.

Minor comments:

The title of this paper is somewhat confusing. Title should be restated with inclusion of keywords and the conclusion of this study.

Thank you very much. Title was changed.

Abstract is too long and hard to read all. Write briefly with including a brief background, purpose, methods, results, and conclusion.

Thank you very much. Abstract of the manuscript was shortened.

What microorganisms were used for fermentation? Include in the M&M.

Dear Reviewer information about microorganisms of Kombucha are added in the manuscript (p.17)

With kind regards,

Authors